# Advancing Alzheimer’s Therapeutics: Exploring the Impact of Physical Exercise in Animal Models and Patients

**DOI:** 10.3390/cells12212531

**Published:** 2023-10-27

**Authors:** Jesús Andrade-Guerrero, Paola Rodríguez-Arellano, Nayeli Barron-Leon, Erika Orta-Salazar, Carlos Ledesma-Alonso, Sofía Díaz-Cintra, Luis O. Soto-Rojas

**Affiliations:** 1Laboratorio de Patogénesis Molecular, Laboratorio 4, Edificio A4, Carrera Médico Cirujano, Facultad de Estudios Superiores Iztacala, Universidad Nacional Autónoma de México, Tlalnepantla 54090, Mexico; jesusandrade1007@gmail.com; 2Departamento de Neurobiología del Desarrollo y Neurofisiología, Instituto de Neurobiología, Universidad Nacional Autónoma de México, Queretaro 76230, Mexico; paola.rdz.inb@gmail.com (P.R.-A.); barronleon1306@gmail.com (N.B.-L.); erikaortabio@yahoo.com.mx (E.O.-S.); neurovet.cla@gmail.com (C.L.-A.); 3Red MEDICI, Carrera Médico Cirujano, Facultad de Estudios Superiores Iztacala, Universidad Nacional Autónoma de México, Tlalnepantla 54090, Mexico

**Keywords:** physical exercise, physical activity, neuroprotection, animal model, Alzheimer’s disease

## Abstract

Alzheimer’s disease (AD) is the main neurodegenerative disorder characterized by several pathophysiological features, including the misfolding of the tau protein and the amyloid beta (Aβ) peptide, neuroinflammation, oxidative stress, synaptic dysfunction, metabolic alterations, and cognitive impairment. These mechanisms collectively contribute to neurodegeneration, necessitating the exploration of therapeutic approaches with multiple targets. Physical exercise has emerged as a promising non-pharmacological intervention for AD, with demonstrated effects on promoting neurogenesis, activating neurotrophic factors, reducing Aβ aggregates, minimizing the formation of neurofibrillary tangles (NFTs), dampening inflammatory processes, mitigating oxidative stress, and improving the functionality of the neurovascular unit (NVU). Overall, the neuroprotective effects of exercise are not singular, but are multi-targets. Numerous studies have investigated physical exercise’s potential in both AD patients and animal models, employing various exercise protocols to elucidate the underlying neurobiological mechanisms and effects. The objective of this review is to analyze the neurological therapeutic effects of these exercise protocols in animal models and compare them with studies conducted in AD patients. By translating findings from different approaches, this review aims to identify opportune, specific, and personalized therapeutic windows, thus advancing research on the use of physical exercise with AD patients.

## 1. Introduction

Alzheimer’s disease (AD) is a neurodegenerative disorder that accounts for approximately 80% of all dementia cases and predominantly affects individuals over the age of 65 [1]. It is characterized by a decline in cognitive functions. The World Health Organization (WHO) estimates that there are currently over 55 million individuals worldwide with dementia, a number that is projected to increase to around 132 million by the year 2050. This escalation presents significant economic challenges and a growing demand for specialized caregiving [2].

Cognitive impairment is a central feature of AD pathology, profoundly impacting various functions such as memory, learning, thinking, behavior, orientation, and judgment. Individuals with AD struggle to retain and recall memories, acquire new information, and process thoughts, leading to a progressive decline in overall cognitive abilities [3]. As a result, AD profoundly impacts the quality of life for both the affected individuals and their caregivers, necessitating comprehensive and compassionate support systems to manage the complex challenges associated with cognitive decline. The progression toward clinical stages of AD involves a transitional phase known as mild cognitive impairment (MCI), which stands between normal aging and dementia. MCI represents a state where cognitive functioning deteriorates more severely than expected for individuals of the same age and educational background [4].

Sporadic AD constitutes 95% of cases [1], and metabolic disorders like type II diabetes, hypertension, vascular issues, diet, sleep disorders, and low physical activity levels are associated with the disease [1,5]. Lifestyle risk factors, including poor diet, obesity, high cholesterol, and sedentary behavior, also contribute to its development [6]. Approximately 1–5% of AD cases exhibit a dominant familial or autosomal pattern. These cases are classified as early-onset and are linked to genetic mutations in presenilin 1, presenilin 2, or the amyloid precursor protein [7].

The pathophysiology of AD is marked by the presence of extracellular plaques composed of amyloid beta (Aβ) peptides in synaptic terminals and blood vessels within the brain. Additionally, intracellular neurofibrillary tangles (NFTs) consisting of hyperphosphorylated tau accumulate in axonal regions, contributing to neuronal loss and synaptic dysfunction [1,8]. In addition to extracellular plaques and NFTs, AD is associated with dysfunction in several neurotransmitter systems. Notably, the cholinergic system is affected by the loss of cholinergic neurons and a decline in choline acetyltransferase activity, not only in late stages, but also in preclinical and early stages [9,10].

Furthermore, the disease process involves the downregulation of proinflammatory cytokines, resulting in heightened inflammation and oxidative stress. Changes in mitochondrial morphology, increased production of reactive oxygen species (ROS), and the presence of bioactive metals like copper, iron, zinc, and magnesium further contribute to the aggregation of Aβ and tau hyperphosphorylation. Alterations in mitochondrial morphology, increased ROS production, and insufficient antioxidant levels collectively promote the escalation of oxidative stress [11,12,13,14].

AD is a pressing global public health concern, and research is key to finding effective treatments. Current treatments for AD primarily focus on improving the patients’ quality of life by managing the symptoms [15]. Given the multifactorial nature of AD, a combination of pharmacological and non-pharmacological approaches is employed to delay and alleviate symptoms, enhancing patient well-being [16]. Physical exercise is one of the non-pharmacological options that is gaining relevance. On the one hand, physical activity is defined as any movement produced by the musculoskeletal system that results in energy expenditure [17]. On the other hand, physical exercise includes planned, dosed, and repetitive activities that seek to enhance or maintain physical fitness. It offers various beneficial effects on overall health and different organs [18]. While physical exercise shows promise as a non-pharmacological approach to alleviate AD symptoms and enhance prevention, its efficacy is hindered by unclear molecular and biochemical mechanisms. Most of the studies selected for the development of this article are from the last 5 years. Concerning animal models, we considered articles that examined the neuroprotector effects of voluntary and forced physical exercise, specifying the dosage and therapeutic outcomes. Regarding human studies, we included those involving physical activity and exercise and those encompassing various ethnicities. This review aims to assess and compare the therapeutic effects of physical exercise in both animal models and AD patients. This article highlights the significance of bridging the gap between findings in animal and human models to enhance our understanding of the neurobiological mechanisms underlying AD and to refine therapeutic approaches.

### 1.1. Beneficial Effects of Various Types of Physical Exercise Used in the Mouse Model

Various murine models enable researchers to utilize and compare diverse types of physical exercise to study their beneficial effects [19,20]. Findings from these models, which are consistent with those of human studies, indicate that physical exercise protects against cognitive decline in AD [21]. Physical exercise in rodents falls into two categories: voluntary and forced. Both types have demonstrated beneficial effects (Figure 1) [22], considering their respective advantages, disadvantages, speed, intensity, and duration (Table 1) [23]. Several types of physical training have been employed in the context of AD, including voluntary wheel running, weighted ladder climbing, swimming, and treadmill running. Each type offers unique advantages for AD research (Figure 1), determined based on its characteristics, recommended usage of equipment, and impact on murine models, as illustrated in Table 1 and Table 2.

In essence, choosing the appropriate exercise type for studies involving AD animal models is a pivotal decision that requires a balanced assessment of their distinct attributes, benefits, and drawbacks. Free wheel running enables long-term research, particularly in neurodegenerative disorders and memory deficits, allowing rodents to engage in self-paced activity and natural behaviors. Environmental enrichment fosters varied stimuli, promoting social interaction and mobility, although challenges in statistical integrity and variable control arise. Forced exercise, exemplified by treadmill usage, facilitates short-term investigations, affording precise intensity control via speed and incline adjustments, yet may entail stress and supervision. Swimming engages whole-body muscles, enhancing memory and alleviating neuropsychiatric symptoms, but tank dimensions and temperature require attention. Resistance training offers strength and cognitive gains, although complexity limits its use. In making this choice, researchers should align objectives with exercise characteristics to deepen our comprehension of exercise’s impact on neuroprotective effects.

Physical exercise interventions have drawn considerable interest for their potential disease-modifying benefits [38]. In the realm of AD animal models, two prevalent approaches—voluntary and forced exercise—have been extensively employed [39], each accompanied by its unique advantages and drawbacks, as elucidated earlier. However, irrespective of the selected method, a robust body of evidence underscores the efficacy of physical exercise as a valuable intervention (summarized in Table 2), encapsulating a spectrum of positive outcomes.

**Table 2 cells-12-02531-t002:** Beneficial effects of voluntary exercise in rodent models of Alzheimer’s disease.

Author/Year	Characteristics of the Subjects	Dosage	Therapeutic Effect
**Voluntary exercise**			
Andrade et al., 2023 [40]	Transgenic 3xTg-AD mice, females, 10 months (*n* = 40).	Voluntary wheel, 2 h a day, 5 days a week for 3 months.	- ↓ Aβ seeding and vascular amyloid. - ↑ Improvement in cognitive behavior and NVU (basement membrane, pericytes, and astrocyte feet).
Wang et al., 2023 [41]	Transgenic APP/PS1 mice, males, 10 months (*n* = 80).	Voluntary wheel with free access for 4 months.	- ↑ Cognitive improvement, dendritic spines, synapses, and AGEs/RAGEs pathway. - ↓ Microgliosis.
Mehla et al., 2022 [42]	Transgenic APP knock-in mice, males, 3 months (*n* = 35).	Voluntary wheel with free access for 9 months.	- ↑ Cognitive improvement. - ↓ Anxiety, Aβ load, and microgliosis. - Preservation of cholinergic cells.
Belaya et al., 2021 [43]	Transgenic 5xFAD mice, male, 1.5 months (*n* = 32).	Voluntary wheel with free access for 6 months.	- ↑ Iron homeostasis and IL6-STAT3-KAK1 pathway. - ↓ IL6.
Liu et al., 2022 [44]	Transgenic APP/PS1 and knock-in AQP-4 mice, males, 3 and 7 months (*n* = 72).	Voluntary wheel, 4 h a day, 5 days a week for 2 months.	- ↓ Aβ seeding. - ↑ Cognitive improvement. - No changes in AQ-4.
Belaya et al., 2020 [45]	Transgenic 5xFAD mice, males, 6 weeks, (*n* = 82).	Voluntary wheel with free access for 6 months.	- Modulates reactive astrocytes, stabilizes anxiety levels, improves spatial memory, and modulates synaptic proteins.
Ziegler-Waldkirch et al., 2018 [46]	Transgenic 5xFAD mice and APP males; 1,5, 4, and 8 months (*n* = 48).	Environmental enrichment and voluntary exercise for 6 weeks.	- ↓ Aβ seeding. - ↑ Survival of neurons (decreases apoptotic cells). - Reverses long-term memory deficit. - Revives adult neurogenesis. - Significantly higher numbers of phagocytic microglia cells.
Do et al., 2018 [47]	Transgenic 3xTgAD mice, male, 3 and 6 months (*n* = 28).	Voluntary wheel with free access for 2 months.	- ↓ Apoptosis, neuroinflammation, Aβ load, and neuronal death. - ↑ POMC and NPY expression; glucose metabolism; cognitive behavior.
**Forced exercise**			
Campos et al., 2023 [48]	Transgenic APP/PS1 mice, males, 6–7 months (*n* = 60).	Climb a ladder with a progressive overload, every other day, for 4 weeks.	- ↓ Aβ seeding. - ↓ Plasma corticosterone levels. - No cognitive changes. - ↑ Numbers of microglial cells in the hippocampus.
Yuan et al., 2022 [49]	Transgenic APP/PS1 mice, male, 3 months (*n* = 24).	Treadmill for 12 weeks 60–70% VO_2_ max, 45 min per session.	- ↑ Cognition, intestinal barrier proteins, gut diversity, and TJ such as ZO-1 and occludin. - ↓ Aβ pathology, neuronal loss, intestinal pathogenic bacteria, microglial and astrocytic activation.
Xu et al., 2022 [50]	Transgenic 3xTg-AD mice, male, 2 months (*n* = 36).	Treadmill for 5 months, 1 h per day, 5 times per week.	- ↓ Aβ seeding. - ↑ Synapses, dendritic spines, and cognitive behavior.
Liu et al., 2022 [51]	Transgenic 2xTg (APPswe/PSEN1dE9) mice, male, 2 months (*n* = 120).	Treadmill and swimming (20 min both), 6 times a week for 4 weeks.	- ↑ Cognitive and exploratory behavior and mitochondrial function. - ↓ TNF-α, GSK3-β, RMP2, P-H2A.X, 8-OHDG, CCO and ATP. - ↓ Anxiety and depression, Aβ42 oligomers, and reactive astrogliosis. - ↑ Modulates synaptic proteins.
Liu et al., 2022 [52]	Transgenic APP/PS1 mice, male, 7 months (*n* = 28).	HIIT 30 min per session for 10 weeks	- ↓ Aβ deposition and astrocytic hypertrophy. - ↑ Cognition and mitochondrial dynamics.
Mu et al., 2022 [53]	Transgenic 3xTg-AD mice, male, 3 months (*n* = 132).	Treadmill 1 h per day, 5 days a week for 12 weeks.	- ↓ GSK3-β activity, Aβ oligomers, ↓ phosphorylation of CRMP2 at Thr514 (preventing axonal degeneration), microglial and astrocyte activation, and IL-1β, IL-6, and TNFα.
Bareiss et al., 2022 [54]	Transgenic 3xTg-AD mice, male, 6 months (*n* = 17).	Forced running wheel, 4 h per day, 3 times a week for 12 weeks.	- Improves sensorimotor symptoms. - Mitigates early deficits in AD.
Liu et al., 2020 [55]	Transgenic 3xTg-AD mice, male, 9 months (*n* = 30).	Ladder climbing exercise, 3 days per week for 12 weeks.	- Improved cognitive performance. - ↓ Amyloid plaques, tau phosphorylation, and microglial and astrocyte activation.
Hashiguchi et al., 2020 [56]	Transgenic APP/PS1 mice, male and female, 6–7 months (*n* = 56).	Treadmill 5 times per week for 1 month.	- Locomotor behavior benefits. - ↓ Amyloid plaques and cytokines levels (IL-1a, IL-4 and IL-6).
Kim et al., 2019 [57]	Transgenic 3xTg-AD mice, male, 3 months (*n* = 30).	Treadmill, 5 days per week for 12 weeks.	- ↓ Aβ burden and neuroinflammatory. - ↑ Mitochondrial function, neurogenesis, and spatial memory.
Wu et al., 2018 [58]	STZ-induced sporadic rat model, male, 2.5 months (*n* = 40).	Swimming exercise, 4 weeks, 1 h per day.	- Improves spatial learning and memory deficits. - ↑ Levels of anti-inflammatory (IL-10) and antioxidant (Nrf2) molecules. - ↓ Aβ 1–42 levels, tau hyperphosphorylation. - ↑ Nrf2 and PARP1 expression.
Lu et al., 2017 [59]	STZ-induced rat model, male, 250–280 g (*n* = 32)	Treadmill for 1 month, 5 times per week, 30 min per day.	- ↑ Cognitive behavior and mitochondrial function. - ↓ Hippocampal neuronal degeneration, Aβ load, tau hyperphosphorylation, oxidative stress, pro-apoptotic proteins, and pro-inflammatory mediators.

Abbreviations: 3xTg-AD: triple-transgenic mouse model of Alzheimer’s disease, 5xFAD: Familiar Alzheimer Disease mice bear five AD-linked mutations model, 8-OHdG: 8-hydroxy-2′-deoxyguanosine, AGEs/RAGEs: advanced glycation end products/receptor for advanced glycation end products, Aβ: amyloid beta, APP/PS1: double transgenic mouse model, AQP-4: Aquaporin 4, ATP: Adenosine triphosphate, CCO: Cytochrome c oxidase, DNA: Deoxyribonucleic acid, GSK-3β: Glycogen synthase kinase-3 beta, GFAP: Glial fibrillary acidic protein, HIIT: high-intensity interval training, IL: interleukin, MAP-2: Microtubule-associated protein 2, Nrf2: Nuclear factor erythroid-2-related factor 2, NPY: Neuropeptide Y, NVU: Neurovascular unit, PARP1: Poly (ADP-ribose) polymerase 1, PSEN1de9: Sequence encoding human presinilin-1 lacking exon 9, P-H2AX: Phospho-histone H2AX, PHF1: PHD finger protein 1, POMC: Proopiomelanocortin, PSD-95: Postsynaptic density protein 95, RMP2: Restricted Moeller–Plesset Second-Order, STZ: Streptozotocin, STAT3: Signal transducer and activator of transcription 3, TJ: tight junctions, TUNEL: Terminal deoxynucleotidyl transferase-mediated dUTP nick-end labeling, TNF-α: Tumor necrosis factor alpha, ZO-1: Zonula occludens-1. ↑ Increase, ↓ Decrease.

The synthesis of diverse exercise interventions in murine models of AD highlights their potential as multifaceted therapeutic strategies. Voluntary exercises, like wheel running and environmental enrichment, demonstrate cognitive improvements and reductions in Aβ seeding but face challenges related to parameter control and hierarchical behavior. Forced exercises, such as treadmills and swimming, exhibit cognitive enhancement, Aβ reduction, and the modulation of neuroinflammatory responses. Both types of exercise offer valuable insights into potential therapeutic strategies for AD. While voluntary exercise captures natural behaviors and preferences, forced exercise provides controlled and precise interventions. Integrating findings from both approaches could yield a more comprehensive understanding of the impact of exercise on cognitive function and AD-related pathologies, aiding in the development of effective interventions for this neurodegenerative disorder.

### 1.2. Exercise as a Treatment for AD Patients

To optimize training procedures, careful consideration should be given to the exercise modality, encompassing its frequency, duration, and intensity. Presently, significant variability characterizes training programs for patients with MCI and AD, often due to their combination with other therapeutic interventions (as seen in Table 3) [15,60,61]. Concerning the training session duration, the prevailing norm centers around sessions lasting 30 to 60 min, spanning one to seven times per week and persisting for 3 to 6 months. Regarding intensity, a diverse spectrum exists, yet moderate intensity remains the prevailing choice [62]. Notably, there is a lack of definitive and precise exercise dosage guidelines tailored for MCI and AD patients.

Studies in patients and animals have shown that physical exercise has multiple targets, and not just one, to exert neuroprotective effects from different angles [63,64]. Nonetheless, it is important to note that the effects of resistance training on the disease are somewhat obscured due to the paucity of exclusive studies in this domain. As a holistic recommendation, a multimodal intervention involving aerobic and resistance exercises, along with flexibility and coordination training, conducted at least two to three times weekly, is advisable as an integral part of treatment [65]. This approach seeks to harness the full potential of exercise-based interventions to benefit AD patients.

**Table 3 cells-12-02531-t003:** Beneficial effects of voluntary exercise in AD and MCI patients.

Author/Year	Characteristics of the Subjects	Dosage	Therapeutic Effect
**Physical activity**			
Park et al., 2023 [66]	Individuals at risk for AD, (*n* = 88,047), >65 years, both genders (Black, Japanese American, Latino, Native Hawaiian, and White).	Moderate and vigorous physical activity in at least the last year (Physical Activity Questionnaire).	- ↓ AD risk. - ↓ Number of deaths from AD.
Liu et al., 2021 [67]	MCI patients (*n* = 57), >60 years, both genders (China).	- Baduanjin (10 postures, breathing, and meditation). - Walking (at 55–75% of HR_max_). - Health education: three sessions/week × 60 min × 6 months.	- Cognitive function improvement. - Modulation of functional connectivity of the DA and NE systems (MRI). - Increased gray matter volume in the right anterior cingulate cortex (MRI).
Kim, 2020 [68]	Mild stage of AD patients (*n* = 35), >65 years, both genders (Korea).	Physical activity, horticultural, musical, and artistic: five sessions/week × 1 h/session × 24 sessions.	- Improved cognitive functions. - Reduced levels of depression.
Venkatraman et al., 2020 [60]	Individuals at risk for AD (*n* = 98), 73 ± 5 years old, both genders (Australia).	Twenty-four months of moderate physical activity, 150 min/week, including daily living activities.	- ↑ WMH (MRI) and strength. - No change in hippocampal volume or Aβ (MRI and PET).
Cox et al., 2019 [69]	MCI patients (*n* = 106), >60 years, both genders (Australia).	Twenty-four months, 150 min/week, moderate physical activity.	- Improved PA levels and strength. - Reduced fat mass and fat distribution.
Pedroso et al., 2018 [70]	Clinically diagnosed AD patients (*n* = 31), >65 years, both genders (Brazil).	Moderate-intensity aerobic exercise: 60 min, three sessions/week, 12 weeks.	- ↑ Amplitude and wave reaction P-300 (EEG).
Law et al., 2018 [71]	Elderly individuals at risk of AD (*n* = 85) average age of 64 years, both genders (USA).	Light, moderate, and vigorous physical activity is measured using a triaxial accelerometer.	- ↓ Aβ 42, total tau, and phosphorylated tau in CFS. - No changes in mild to vigorous physical activity biomarkers.
**Physical exercise**			
de Farias et al., 2021 [72]	Clinically diagnosed AD patients (*n* = 15), 68.3 ± 13.8 years, women, (Brazil).	Twenty-two sessions of 60 min aerobic exercise, twice weekly, at 40–60% of HR_max_.	- Improved judgment and problem-solving. - ↑ BDNF and IL-4 anti-inflammatory markers. - ↓ ROS and neuronal damage markers.
Puente-Gonzalez et al., 2021 [73]	Clinically diagnosed AD patients (*n* = 72) >50 years, both genders (Salamanca, Spain).	Multimodal exercise: 50 min, three sessions/week, 6 months (aerobic, resistance, stretching).	- ↑ Bone mineral density. - Better physical capacity. - ↓ Risk of falls.
Vidoni, 2021 [74]	Individuals at risk for AD (*n* = 117), >65 years, both genders (White, African American).	Three to five sessions/week, max 50 min/session, 150 min moderate-intensity aerobic exercise/week, 52 weeks.	- ↑ Executive, verbal, and visuospatial functions. - No changes in hippocampal volume (MRI) and Aβ (PET).
Enette et al., 2020 [75]	Clinically diagnosed AD patients (*n* = 52), >65 years, both genders (France).	Two sessions/week, 30 min/session, 9 weeks, 50–70% of HR_max_.	- ↑ BDNF in blood plasma. - Improvement in physical condition and functional capabilities.
Pedrinolla et al., 2020 [76]	Clinically diagnosed AD patients (*n* = 39), >60 years, both genders (Italy).	Moderate-intensity aerobic and resistance exercise: 90 min, three sessions/week, 6 months.	- ↑ VEGF in blood plasma. - ↑ Brachial and femoral dilation, blood flow, and VO_2_.
Broadhouse et al., 2020 [77]	Clinically diagnosed dementia (*n* = 100), >55 years, women (Australia).	Resistance exercise: 90 min, two to three times/week, 18 months (80% of HR_max_).	- ↑ Cognitive functions and neuroplasticity via EEG. - ↓ Hippocampal atrophy (MRI).
Chang et al., 2020 [78]	Clinically diagnosed AD patients (*n* = 40), 79.3 ± 5.1 years, women. *	Resistance exercise: 40 min, three sessions/week, 12 weeks.	- Improved symptoms of depression. - ↑ Maximum strength. - No change in hypertrophy.
Jensen et al., 2019 [79]	Clinically diagnosed AD patients (*n* = 198), >60 years, both genders. *	Three sessions/week, 60 min/session, 16 weeks, moderate to high intensity (treadmill, stationary bike).	- Modulation of inflammation.
van der Kleij et al., 2018 [80]	Patients with mild to moderate AD moderate stage (*n* = 51), 60–90 years, both genders. *	Moderate intensity aerobic exercise: 16 weeks, 60 min, three sessions/week, (70–80% VO_2_ max).	- No change in VO_2_ or cerebral blood flow (MRI).
Schultz et al., 2015 [81]	Patients at risk of AD (*n* = 69), average age of 64 years, both genders, USA.	Aerobic exercise (graded exercise testing; GXT)	- VO_2_ peak was associated with improvement in immediate memory and verbal learning and reduced Aβ burden (PiB-PET).

Abbreviations: Aβ, amyloid beta; AD, Alzheimer’s disease; BDNF, brain-derived neurotrophic factor; CSF, cerebrospinal fluid; DA, dopamine; EEG, electroencephalogram; HRmax, maximum heart rate capacity; IGF-1, insulin-like growth factor 1; IL, interleukin; MCI, mild cognitive impairment; MEG, magnetoencephalography; MRI, magnetic resonance imaging; NE, norepinephrine; PA, physical activity; PET, positron emission tomography; PiB-PET, Pittsburgh Compound B-positron emission tomography; ROS, reactive oxygen species; sTREM2, soluble triggering receptor expressed on myeloid cells 2; VEGF, vascular endothelial growth factor; VO_2_ peak, maximal oxygen consumption; WMH, white matter hyperintensities. ↑ Increase, ↓ decrease. * Country is not specified in the study.

The body of evidence presented in these studies (Table 3) emphasizes the potential of physical activity and exercise interventions in addressing cognitive decline associated with AD. From aerobic exercise to resistance exercise, a range of modalities demonstrated diverse cognitive benefits and biomarker modifications, suggesting a multifaceted approach to enhancing cognitive function in AD patients. These findings underscore the importance of tailored exercise interventions as a promising avenue in the pursuit of effective AD management strategies. Further research and well-structured clinical trials are warranted to unravel the precise mechanisms underlying these effects and to provide evidence-based recommendations for optimizing exercise interventions for individuals at risk or diagnosed with AD and MCI.

### 1.3. Beneficial Mechanisms of Physical Exercise in Alzheimer’s Disease

Numerous studies conducted on mouse AD models have consistently highlighted the positive effects of exercise and physical activity on memory-related tasks. This correlation extends to individuals diagnosed with AD, where engaging in exercise and physical activity has demonstrated tangible enhancements in mental well-being, cognitive progress, and brain functionality (Figure 2) [82,83,84,85].

Exercise triggers a cascade of cellular and molecular transformations within the mouse brain. These cascades facilitate a range of physiological phenomena, including the promotion of neurogenesis and the activation of neurotrophic factors. These factors, such as those contributing to long-term potentiation (LTP), play pivotal roles in bolstering learning, memory, and neural plasticity. Moreover, exercise contributes to a reduction in Aβ peptides, minimizes the formation of NFTs, dampens inflammatory processes, and mitigates oxidative stress (as depicted in Figure 1 and Figure 2) [86,87]. Therefore, physical exercise exerts neuroprotective effects by acting on multiple targets, rather than relying on a single mechanism.

#### 1.3.1. Beneficial Mechanisms of Physical Exercise on Neuropathological Hallmarks of AD

The core neuropathological characteristics of AD encompass Aβ peptide and the hyperphosphorylated tau protein. Consequently, most of the research pertaining to the influence of exercise has been centered on assessing its impact on these defining attributes. In multiple investigations employing murine disease models, interventions involving both mandatory and voluntary exercise have consistently demonstrated a dampening effect on these traits [88,89,90,91]. Several potential mechanisms have emerged, encompassing diminished Aβ production, augmented Aβ clearance, and reduced NFTs (Figure 2) [92,93].

Patient studies employ specialized tools like positron emission tomography (PET) radiotracers, including the Pittsburgh B (PiB) compound, and magnetic resonance imaging (MRI) to effectively identify Aβ peptide deposits and their neurological consequences. These techniques also allow for a precise assessment of how exercise impacts these pathological markers. By using 11C-PiB PET, researchers obtain images of Aβ aggregates and measure the cerebral quantity of Aβ [94,95,96]. Notably, these studies highlight that engaging in physical activity could potentially enhance the clearance of Aβ 1–42 or reduce its deposition [97,98]. However, to thoroughly understand the effects of exercise on both Aβ and tau imaging, rigorous clinical trials are essential. These trials would bridge the gap between experimental findings and potential applications for AD diagnosis and treatment.

Animal studies have demonstrated that interventions involving forced and voluntary exercise can attenuate neuropathological signs [88,89,90,91]. Concerning Aβ peptide levels, physical exercise appears to influence volume modulation, particularly for Aβ 1–42, in brain regions like the hippocampus and neocortex [99]. In models of tauopathy, exercise has been shown to reduce brain tau phosphorylation [100]. This cumulative body of evidence suggests that exercise could hold promise as a therapeutic strategy to address the neuropathological aspects of AD.

#### 1.3.2. Impact of Physical Exercise on Neurotrophins

Neurotrophic factors (NTFs) and their receptors play a crucial role in neural cell maturation and proliferation, regulate the development and survival of neurons, and appear to be involved in the endogenous neuroprotection of different neurons [101]. NTFs support neuronal survival and function in the adult central nervous system (CNS), generating broad interest in the use of these factors to intervene in neurodegenerative diseases [102].

Most of the favorable effects of exercise in AD have been ascribed to signaling enhancement and the release of NTFs, notably brain-derived neurotrophic factor (BDNF), nerve growth factor (NGF), glia-derived neurotrophic factor (GDNF), and type 1 insulin-like growth factor (IGF-1). These factors are essential in maintaining neuronal functionality and fostering neuroplasticity (Figure 2) [103]. Decreased levels of BDNF in blood plasma have been linked to neurodegeneration, a decline in hippocampal volume, and cognitive impairment associated with AD [103]. Nonetheless, BDNF activates an array of signaling pathways that are integral to neural function, and its levels have a positive impact on memory and a reduction in cognitive decline, as evidenced through exercise interventions [103,104,105,106].

On the other hand, NGF plays a pivotal role in ensuring the survival of cholinergic neurons and shielding them against chemical stressors. In the context of AD, a reduction in NGF has been linked to cognitive deterioration, while exercise in AD mouse models has demonstrated its potential to enhance the expression of this factor. Nevertheless, exercise has inconsistent effects on NGF levels in patients [103]. Conversely, diminished levels of IGF-1 have been associated with both aging and AD [107,108]. However, a rat model of AD exhibited elevated serum IGF-1 levels and improved working memory following aerobic and resistance exercise interventions, suggesting a positive impact on cognition [108]. Similarly, in patients with AD, exercise has the capacity to elevate IGF-1 levels, thereby potentially yielding enduring effects [107]. This collective understanding underscores exercise’s multifaceted impact on essential NTFs, offering a promising avenue for addressing cognitive impairments in AD.

#### 1.3.3. Influence of Physical Exercise on Neuroinflammation

Neuroinflammation is an inflammatory reaction in the CNS that includes immune cell infiltration, microglial activation, and pro-inflammatory cytokine release (Figure 2) [109]. In postmortem AD brains, inflammatory markers have been identified surrounding the extracellular Aβ deposits [110]. In murine models of AD, exercise has a downregulating effect on proinflammatory cytokines such as IL-1β, IL-6, and TNF-α, contributing to a less inflammatory environment [111,112]. Moreover, exercise triggers notable morphological and functional transformations in astrocytes by promoting their growth and attenuating astrogliosis. This suggests that physical activity can potentially modulate inflammatory responses within the brain, particularly in the hippocampus [20,113]. Cognitive improvements induced by exercise in mice are associated with the modulation of neuroinflammation [114].

Likewise, in individuals with AD, elevated levels of cytokine profiles, such as IL-6, TNF-α, and IL-1β, have been seen in plasma and cerebrospinal fluid samples [79,115]. These cytokines are known to stem from reactive astrocytes and microglia, as observed in postmortem AD patient brains [116,117,118], and impact gliotransmission and neurotransmitter uptake, which are closely associated with cognitive dysfunction [119]. Notably, aerobic exercise can reduce elevated cytokine levels [120,121], suggesting that it can potentially counteract the inflammation-related cognitive decline observed in AD.

#### 1.3.4. Effects of Exercise on Oxidative Stress

In AD, an elevation in ROS and reactive nitrogen species (RNS) production leads to oxidative stress, exacerbating damage linked to inflammatory responses and instigating neurodegeneration through apoptosis [122]. Interestingly, physical exercise emerges as a significant contributor to neuronal activation in the hippocampal region, as it requires a heightened mitochondrial capacity to generate ATP via the oxidative phosphorylation of glucose. Paradoxically, this heightened oxidative activity also triggers ROS accumulation, posing a threat to neurons [123]. However, exercise initiates a cascade of counteractive mechanisms that enhance mitochondrial function and attenuate the impact of ROS (Figure 2) [124]. Enzymes like superoxide dismutase 1 and 2 (SOD1 and SOD2) and increased catalase levels, elevated through exercise, augment brain antioxidant capacity [89,90,108]. Notably, chronic exercise in AD mouse models boosts glutathione (GSH), the primary antioxidant enzyme, along with the essential tripeptide (glutamate, cysteine, and glycine) [108,125].

In AD, elevated lipid peroxidation contributes to neuronal membrane damage. Malondialdehyde is a significant marker of this deterioration, but aerobic exercise, resistance exercise, and forced swimming interventions have demonstrated the ability to diminish it [108]. Patient studies substantiate the efficacy of exercise in decreasing pro-oxidant parameters and bolstering antioxidant capacity, irrespective of exercise type or intensity. Exercise has also been linked to reduced ROS levels and catalase serum activity, accompanied by heightened nitrite levels and interleukin 4 (IL4), affirming its potency in curbing oxidative stress in AD [72].

Collectively, the interplay between oxidative stress and physical exercise in AD underscores the potential of exercise interventions to strategically counteract the detrimental effects of ROS and RNS production. These insights not only shed light on the multifaceted nature of AD pathogenesis but also present a compelling case for the inclusion of exercise as a promising adjunctive therapeutic strategy for attenuating the oxidative burden and potentially slowing the progression of AD.

#### 1.3.5. Effects of Exercise on Neurotransmitters

Neuropsychiatric symptoms during AD-like aggression, agitation, and depression are attributed to an alteration in neurotransmission systems, including the serotoninergic, dopaminergic, noradrenergic, and cholinergic systems [126]. Cholinergic system alterations are present in the early stages of the disease and are exacerbated with degeneration [14,127]. In the Veronese study, it is proposed that exercise modulates neurotransmitter production, thereby decreasing depressive behavior and aggression [126]. It has been found that voluntary and forced physical exercise prevent the loss of cholinergic innervation to the hippocampus as well as the decrease in cholinergic fibers, resulting in improved cognition and motor skills [128].

Serotonergic system activation through physical exercise benefits cognitive performance and the emotional component. Animal studies using physical exercise have reported an increase in the activity of serotonergic neurons. This effect has been seen with both acute and chronic exercise [128].

Another neurotransmission system that benefits from the practice of exercise is the dopaminergic system. Dopamine levels rise in the hippocampus after exercise. Moreover, in an animal study with an aerobic exercise intervention, such as swimming, researchers found increased dopamine levels that were associated with memory improvement [128].

Regarding norepinephrine (NE), studies have shown that long-term exercise may lead to increased levels of NE [128]. Voluntary exercise in mouse models increases NE levels in the hippocampus and amygdala, which is proposed to contribute to the control of neuropsychiatric symptoms [126].

In patients, exercise intervention can increase the availability of tryptophan, a serotonin precursor that can decrease depressive symptoms and improve cognition [126,128]. Concerning dopamine, receptor affinity is enhanced, and dopamine levels increase in the hippocampus [38,128]. In patients with MCI, aerobic exercise has been shown to increase NE levels and improve memory [129]. However, there are no data regarding the effects on acetylcholine levels in patients with this type of intervention.

The relationship between exercise and neurotransmission systems presents a hopeful avenue for enhancing cognitive function and managing neuropsychiatric symptoms in AD. Further exploration through clinical trials and comprehensive studies will be essential to fully understand the potential of exercise as a therapeutic strategy for alleviating the cognitive and emotional burdens associated with the disease.

#### 1.3.6. Effects of Exercise on Neurogenic and Anatomical Aspects

In their seminal studies, Van Praag and colleagues (1999) [130] established the existence of neurogenic zones in the hippocampus of adult mice. These zones are situated in the subventricular and subgranular regions of the dentate gyrus [131]. In both regions, voluntary wheel running has been linked to an augmentation in the maturation of new neurons, complementing their proliferation, survival, and differentiation into new entities [132,133,134,135].

Aerobic exercise and physical activity also attenuate brain atrophy, which is correlated with gray matter volume and white matter diffusion tensor imaging, as well as with cognitive severity [136,137,138]. This effect is rooted in the capacity of exercise to induce plasticity and functional changes in the brain. Similar structural modifications have been demonstrated in mouse models of AD, where both enriched environments and physical exercise interventions induced distinct structural changes within regions like the cerebellum, cerebral cortex, and hippocampus when compared to sedentary groups [139,140].

Human studies involving histological markers and cellular division assessments via BrDU or Carbon 14 have further validated hippocampal neurogenesis in adults [141,142]. Additionally, the concept of separation patterns, denoting the differentiation of similar experiences via distinct activity patterns, has been associated with the observation of neurogenesis in the hippocampus. This correlation is supported by studies of young adult participants who underwent light physical exercise interventions. These studies revealed memory enhancement through the mediation of separation patterns within the dentate gyrus [141,143].

The integration of these insights into clinical practice holds tremendous potential. Incorporating exercise interventions as a proactive approach to stimulate neurogenesis and combat brain atrophy could pave the way for novel strategies in treating neurodegenerative conditions, including AD (Figure 2). By harnessing the power of exercise and leveraging these findings to promote neuroplasticity and structural changes, scientists can substantially improve cognitive health and enhance quality of life.

#### 1.3.7. Effects of Exercise on Cognition

Considering that AD predominantly manifests with cognitive symptoms rather than motor deficits, it is imperative to comprehensively assess the impact of exercise on cognition. Thus, a strong body of evidence has emerged from animal models [135]. Studies in AD mouse models have evaluated a range of parameters, including learning, spatial memory, working memory, exploratory behavior, and affective behavior. Although some results have been inconsistent [144,145], possibly due to the diversity of the utilized models, variations in exercise protocols, and the spectrum of behaviors assessed, most studies agree that physical exercise has a positive impact and no detrimental effects on cognitive function have been observed [146,147].

In studies involving human patients, the findings remain heterogeneous. A systematic review covering 13 studies with a total of 869 AD patients unveiled significant effects in eight studies, while five reported no discernible changes following interventions [148]. Another comprehensive report encompassing 1145 patients with MCI and AD revealed favorable outcomes attributed to aerobic exercise [149]. Subsequently, a study by Demurtas et al. (2020), encompassing 14,209 patients with MCI and dementia, highlighted positive exercise effects on global cognition, although no disparities were noted in specific evaluations of attention, executive function, and memory [150].

Consequently, discerning the precise impact of exercise on cognition is complex due to the dearth of specific exercise guidelines for brain health and AD. This complexity arises from the diverse array of protocols applied across different disease stages as well as the heterogeneous assessment methods and scales employed, all of which contribute to the variances in outcomes. As such, substantiating the benefits of exercise on cognitive functions in the context of AD requires a more robust body of evidence.

#### 1.3.8. Effects of Exercise on the Neurovascular Unit

Aβ peptide deposition is not exclusive to neuritic plaques within the brain parenchyma. It also accumulates in cerebral blood vessels, leading to cerebral amyloid angiopathy (CAA) [151]. The histopathological transformations observed in CAA are intimately interconnected with the neurovascular unit (NVU) and blood–brain barrier (BBB) dysfunction. In AD, the integrity of both the BBB and cerebral blood flow is compromised due to NVU components, such as pericyte degeneration, endothelial cell alterations, astrocytic foot dysfunction, and basal membrane deterioration, which collectively exacerbate disease progression [152]. Although specific treatments for vascular dysfunctions are lacking, exercise emerges as a potential therapeutic measure.

Evidence from the TgCRND8 model of vascular amyloidosis indicates that exercise intervention yields reductions in the CAA along with hippocampal vasculature normalization [153]. Similarly, our working group demonstrated CAA reduction after three months of voluntary exercise, coupled with beneficial effects on vascular morphology and NVU components in the 3xTg-AD model (Figure 2) [40]. Another study employing the 5xFAD model reported that exercise can ameliorate BBB dysregulation, fostering pericyte proliferation and elevating levels of ZO-1 and claudin-5 proteins over a four-month treatment period [154].

While human research into the effects of exercise on these alterations remains limited, increased cerebral blood flow, reduced cardiovascular risk factors, and enhanced angiogenic factors following interventions have been linked to improved memory [76,155,156]. These findings collectively underscore the pivotal role of physical exercise in addressing vascular disorders, presenting a promising avenue for comprehensive treatment strategies.

#### 1.3.9. Effects of Exercise on Metabolism

AD is closely interlinked with metabolic diseases and their alterations. An evident association is observed in the intricate connection between type 2 diabetes mellitus (T2DM) and cognitive impairment. Furthermore, AD is distinguished by a reduction in glucose consumption, directly impacting learning and memory capacities. Perturbations in glucose metabolism reverberate through neurotransmission maintenance and neuronal function, prompting considerable investigation into the link between AD and metabolic pathways, including enzymes associated with glycolysis [157,158].

Lifestyle habits play a pivotal role in the context of AD. Obesity stands as a prominent risk factor due to its potential to induce insulin resistance, intricately intertwined with AD. Notably, dietary habits come into play; individuals consuming diets that are rich in cholesterol, saturated fats, and hypercaloric components exhibit an elevated AD risk, while those favoring fiber, vegetables, and fruits demonstrate lower susceptibility [157,158].

In this landscape, regular exercise emerges as a potent tool for overall health, capable of preventing and managing various health conditions. Exercise serves as a preventive measure and treatment approach for obesity and metabolic dysregulation. Its impact extends to positively influencing metabolic syndrome, obesity, insulin resistance/T2DM, dyslipidemia, and hypertension [159,160].

Moreover, physical exercise fosters glucose uptake by facilitating crucial processes such as efficient glucose delivery, transportation across muscle membranes, and heightened intracellular flux through metabolic pathways involved in glycolysis and glucose oxidation [161,162]. Consequently, exercise assumes a pivotal role in the prevention and management of overweight and obesity. The WHO recommends 150 to 250 min of moderate-intensity physical exercise per week to stave off weight gain. A plethora of studies underscore exercise’s capacity to enhance cardiometabolic health, insulin sensitivity, and lipolysis [159].

Looking ahead, integrating exercise interventions as a comprehensive strategy for managing and potentially preventing AD holds promise. However, challenges remain, including the need for tailored exercise guidelines and understanding the optimal exercise regimens for various stages of the disease. Long-term studies examining the sustained impact of exercise on metabolic and cognitive health in AD patients are essential for providing solid evidence for its effectiveness.

### 1.4. Dosage and Recommendations for Physical Exercise in Patients with AD

Currently, a significant degree of variability persists in the training programs designed for individuals with AD. This variability stems from the amalgamation of exercise with other therapeutic interventions, showcasing the multifaceted nature of interventions targeting this condition [15,61,62]. In terms of session duration, the majority of programs emphasize interventions lasting from 30 to 60 min, conducted between one and seven times per week, with the protocols generally spanning 3 to 6 months. The intensity levels exhibit a broad spectrum, spanning from mild to vigorous, with moderate intensity emerging as the most frequently utilized [62]. This inherent diversity has led to the absence of specific exercise dosage guidelines tailored to AD.

Among the interventions explored, aerobic exercise has taken precedence in both patient and animal model studies, with resistance exercise comparatively receiving less attention. Nonetheless, positive outcomes have been documented, particularly in enhancing functionality and overall quality of life among patients. To assist in navigating the realm of physical activity and exercise for AD, Table 4 outlines some essential dosage guidelines and recommendations. It becomes evident that a standardized approach to exercise prescription for this population remains a challenge, warranting further research and the establishment of comprehensive guidelines.

## 2. Conclusions and Perspectives

This discussion highlights the significance of applying exercise interventions in AD, transitioning from animal models to human subjects. While animal studies reveal cognitive enhancements and biomarker changes through exercise, translating these findings to clinical practice in humans is complex due to inter-species differences and the multifactorial nature of the disease. Nevertheless, the cognitive benefits, reduction in neuroinflammation, mitigation of oxidative stress, and modulation of synaptic proteins observed in animal models and human trials support the multi-target effects of exercise interventions. To therapeutically translate these insights, a comprehensive approach encompassing both aerobic and resistance exercise, tailored to individual capacities and disease stages, is crucial. Multimodal interventions, considering aspects like flexibility and nutrition, offer potential synergistic effects. Despite the challenges, a personalized approach that draws inspiration from animal models while accommodating human nuances holds the potential for effective AD interventions. Interdisciplinary research and well-designed trials are needed to optimize exercise strategies for AD patients.

The studies examined in this review focus on highlighting the beneficial effects and mechanisms of exercise in AD. This is demonstrated through various training protocols in both animal models and patients at different disease stages, covering morphological, behavioral, molecular, and physiological aspects. The aim is to identify opportunities for further research and propose optimal applications of exercise in AD treatment.

Although evidence in animal models confirms the safety and benefits of exercise in AD, more clinical data are required regarding its impact on neuropathological aspects of the disease, as the current information is limited. Thus, it is imperative to conduct studies with rigorous controls, targeting specific stages and employing diverse protocols to establish efficient types and dosages of exercise, culminating in the formulation of precise treatment guidelines.

The main limitation in clinical trials involving AD patients is the lack of established exercise dosages customized to the patient’s clinical stage, followed by an examination of the methodological limitations related to exercise recommendations for individuals with AD, depending on their clinical stage. Another gap is the lack of global health promotion and prevention campaigns that highlight the neuroprotective role of exercise at various stages of life, especially in youth, since AD develops decades before cognitive symptoms appear. Therefore, clinical studies and future researchers should be directed toward evaluating the effects of exercise with different protocols in healthy individuals and several stages of AD patients.

Interpreting findings between animal and human studies poses challenges due to discrepancies in exercise protocols and the wide range of parameters, sampling methods, and results. However, this presents an opportunity to delve into the underlying mechanisms through which exercise affects AD. Moreover, the advantages of exercise remain evident due to its positive impact on overall health and well-being, which involves intricate interactions between the brain and the body. Promoting lifestyle modifications, particularly during the pre-symptomatic and predementia stages, has the potential to mitigate disease progression, enhance cognition, delay neuropsychiatric symptoms, and improve patients’ quality of life, all while reducing palliative care costs. Therefore, concerted efforts are required to advance the infrastructure and knowledge related to physical exercise interventions in the healthcare sector.

## Figures and Tables

**Figure 1 cells-12-02531-f001:**
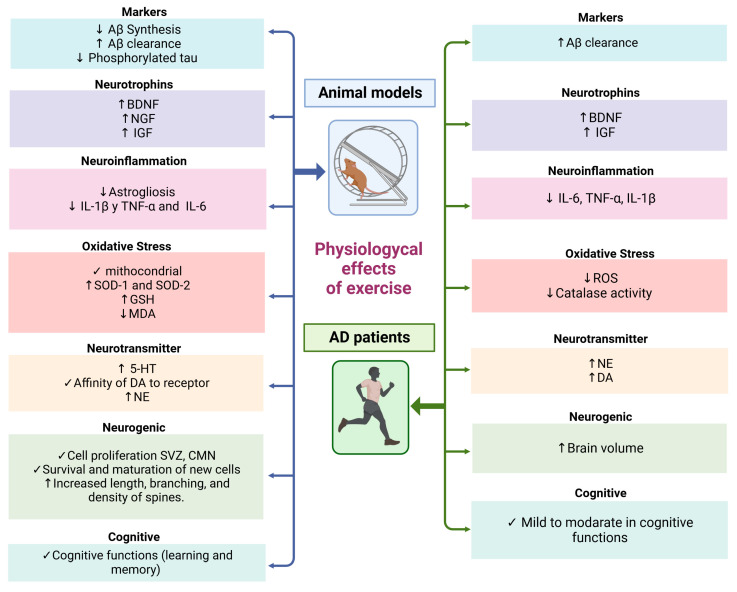
Multiple neuroprotective targets of physical exercise in AD patients and animal models. Abbreviations—Aβ: amyloid beta; AD: Alzheimer’s disease; BDNF: brain-derived neurotrophic factor; IGF: insulin-like growth factor; NGF: nerve growth factor; ROS: reactive oxygen species; IL-6: interleukin 6; IL-1β: interleukin-1β; TNF-α: tumor necrosis factor alpha; IL-10: interleukin 10; 5-HT: 5-hydroxytryptamine; DA: dopamine; NE: norepinephrine; SVZ: subventricular zone; CMN: congenital mesoblastic nephroma; SOD-1: superoxide dismutase 1; SOD-2: superoxide dismutase 2; GSH: glutathione; MDA: malondialdehyde. ↑ (Increase) ↓ (Decrease) ✓ (Improvement). This figure was created with BioRender.com (accessed on 13 October 2023).

**Figure 2 cells-12-02531-f002:**
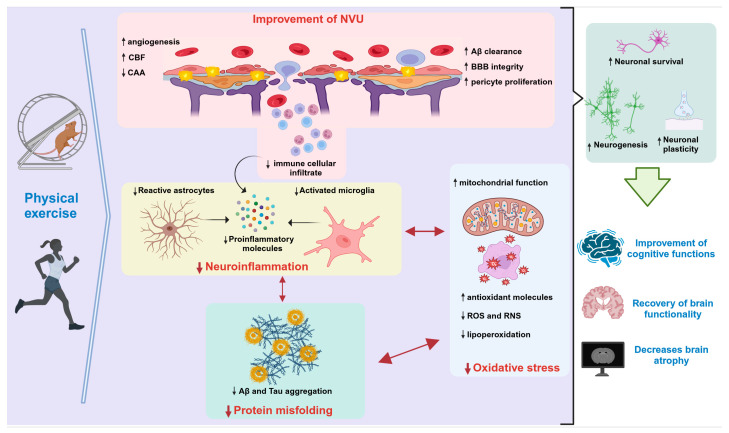
Neuroprotective effects of physical exercise in AD. Abbreviations: Aβ: amyloid beta; BBB: blood–brain barrier; CAA: cerebral amyloid angiopathy; CBF: cerebral blood flow; NVU: neurovascular unit; RNS: reactive nitrogen species; ROS: reactive oxygen species. ↑ (Increase) ↓ (Decrease). This figure was created with BioRender.com (accessed on 13 October 2023).

**Table 1 cells-12-02531-t001:** Types of voluntary and forced exercises used with mouse models of Alzheimer’s disease.

Exercise Type	Characteristics	Advantages	Disadvantages
**Voluntary exercise**
Wheel running 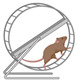	-Long-term studies, even lifelong studies.-Recommended for neurodegenerative diseases and memory deficits [24].	-Rodents can run freely at a self-selected intensity.-Natural behavior and playing activities.-Does not trigger stress [23].	-Only for small-sized rodents.-Monitoring of activity is recommended to control the amount of exercise performed [25].
Environmental enrichment 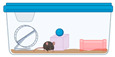	-Enrichment objects differ in composition, shape, size, texture, smell, and color, including cages with toys, nesting material, and hiding tubes [26].	-Offers enhanced social interaction and movement freedom, combining social and inanimate stimuli.-Provides diverse cognitive, sensory, social, visual, somatosensory, and olfactory stimulation, with added cognitive focus on spatial map formation [26].	-Potential harm to animals, statistical influences, data interpretation challenges, and hindered replication.-Hierarchical and territorial behavior in social animals can impact both high- and low-ranking individuals.-Incomplete control over parameters and independent variables [27,28].
**Forced exercise**
Treadmill 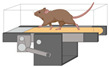	-Commonly used in short-term studies (6–12 weeks) and 10–30 min per day.-Exercise capacity is defined by 4 parameters: VO_2_ max, maximum blood lactate, steady state, and maximum heart rate [29].	-Dosage control is achievable through speed, incline, and training duration adjustments.	-A 3 to 10-day adaptation period is necessary before protocol initiation.-Manipulation and stress conditions.-Supervision by one or more experimenters is required.-Expensive equipment.-Sex- and strain-related variations mandate intensity adjustments for optimal training performance [23,30].
Swimming 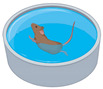	-Mostly used in aerobic protocols [31].-Recruit muscles and ligaments of the whole body.	-Swimming enhances short- and long-term memory and alleviates neuropsychiatric symptoms [32].-Simple equipment (tank).	-The tank should be big and deep enough for the animal’s size.-The temperature should be at 30–32 °C-Potentially more stressful than alternative exercise forms [23,30].
Resistance training/ladder 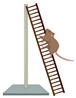	-Weighted ladder is mainly used. Intensity is determined by repetitions and the training phase.-Protocols use stimulation sets, 3 min rest, and 80–85% ladder slope [23,29].	-Enhances strength, hypertrophy, and cognitive function.-Yields substantial muscle hypertrophy, unlike other exercises [33,34,35].	-External aids (electrical stimuli, surgery, equipment) are crucial for animal exercise.-Complex manipulation limits common use.-Muscle hypertrophy varies by muscle type [23,36,37].

Abbreviations: VO_2_ max, maximum oxygen consumption.

**Table 4 cells-12-02531-t004:** General recommendations and proposed exercise dosage for AD patients.

Proposed Dosage for AD Patients			
	Frequency per Week	Time in Minutes	Intensity
Physical activity	5 times	30 min	Moderate (3–6 METS)
**Physical exercise**			
Strength and resistance	2–3 times	20–30 min	Moderate to vigorous (RM or Borg scale 6–7)
Aerobic exercise	3 times	35–50 min	Moderate (50–80% VO_2_ max or 4–7 RPE)
Coordination and balance	3 times	15 min	Mild to moderate (4–6 RPE)
Stretching	3 times	10 min	Low (2–3 RPE)
**General recommendations:**			
Conduct pre-training physical examinations for cardiovascular risk patients. Guide patients who are unable to follow instructions by mimicking activities. Emphasize familiar, polyarticular movements for patient engagement. Change environmental and social factors for varied training. Combine multimodal training with frequent cognitive stimulation. Learn training to guide patients at home.			

Abbreviations: METS, metabolic equivalent of the task; RPE, rate of perceived exertion on Borg scale; RM, repetition maximum; VO_2_ max, maximal oxygen consumption. These recommendations were obtained following what is described in [73,163,164,165].

## Data Availability

The data that support the findings of this review are available from the corresponding author upon reasonable request.

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
