# Peer review of "Advancing Alzheimer’s Therapeutics: Exploring the Impact of Physical Exercise in Animal Models and Patients"

_cells, 2023, doi:10.3390/cells12212531_

Round 1

Reviewer 1 Report

Comments and Suggestions for Authors

Reading this review on assessing and comparing the therapeutic effects of physical exercise in both animal models and AD patients was a pleasure. Overall, the manuscript is well written and will be useful for the reader of Cells.

I have the following comments to improve the quality of the manuscript.

1. In line 86 change the word “study” into “review”.

2. Add some lines about the methods of how articles were selected to be used for this review.

3. The sentence in Lines 87-89 is quite general however, the manuscript focuses on therapeutic effects of physical exercise in both animal models and AD patients, so please change it accordingly.

4. Please mention the following article in your review “Campos HC, Ribeiro DE, Hashiguchi D, Glaser T, Milanis MdS, Gimenes C, Suchecki D, Arida RM, Ulrich H and Longo BM (2023) Neuroprotective effects of resistance physical exercise on the APP/PS1 mouse model of 

Alzheimer’s disease. Front. Neurosci. 17:1132825.doi: 10.3389/fnins.2023.1132825”.

5. If possible, highlight all the literature referred from human studies by stating it.

6. Line 474-475 – please add the references for table 4.

7. Add DOI for references 17, 27, 40, 90

Author Response

Reading this review on assessing and comparing the therapeutic effects of physical exercise in both animal models and AD patients was a pleasure. Overall, the manuscript is well written and will be useful for the reader of Cells.

I have the following comments to improve the quality of the manuscript.

We thank the reviewer for the effort in analyzing our manuscript. We have highlighted in yellow the changes and suggestions made by the reviewer.

Q1. In line 86 change the word “study” to “review”.

A1. The change has been made.

Q2. Add some lines about the methods of how articles were selected to be used for this review.

A2. In the introduction section we have described how the articles to be used for this review were selected.

Q3. The sentence in Lines 87-89 is quite general however, the manuscript focuses on therapeutic effects of physical exercise in both animal models and AD patients, so please change it accordingly.

A3. The suggestion made by the reviewer has been implemented.

Q4. Please mention the following article in your review “Campos HC, Ribeiro DE, Hashiguchi D, Glaser T, Milanis MdS, Gimenes C, Suchecki D, Arida RM, Ulrich H and Longo BM (2023) Neuroprotective effects of resistance physical exercise on the APP/PS1 mouse model of Alzheimer’s disease. Front. Neurosci. 17:1132825.doi: 10.3389/fnins.2023.1132825”.

A4. We appreciate the reviewer for the suggestion and we have added the article in Table 2.

Q5. If possible, highlight all the literature referred from human studies by stating it.

A5. Literature related to human studies is in section “1.2. Exercise as a treatment for AD patients” and we have verified that it is stated in section 1.3.

Q6. Line 474-475 – please add the references for table 4.

A6. We have added the references used in Table 4.

Q7. Add DOI for references 17, 27, 40, 90

A7. We appreciate the reviewer's observation, however, we have used the endnote and the style requested by the journal

Reviewer 2 Report

Comments and Suggestions for Authors

The Paper entitled “Advancing Alzheimer's Therapeutics: Exploring the Impact of

Physical Exercise in Animal Models and Patients by Jesús Andrade-Guerrero and co-authors highlighted the beneficial effects of exercise on Alzheimer’s disease-related pathologies in animals and humans. They have covered the underlying AD-related pathologies, such as oxidative stress, neuroinflammation, and cognitive functions. The paper has been strengthened with several tables and graphical illustrations. The overall flow of the paper is simple, stable, and nicely designed.

1.       The most part of the abstract is related to AD-pathology, it must present a high proportion of the exercise-inducing effects.

2.       The given conclusion is insufficient, more research gaps may be targeted and highlighted, which may be a clue for future researchers.

3.       The paper is quite lengthy, efforts may be made to summarize the main things and make it easier for the authors to make it publishable.

4.       Highlight the shortcomings or limitations of the current research on the role of exercise in preventing or managing AD-related pathologies, which may be added in the conclusion and future perspectives.

5.       Figure-1, effects of exercise on animal models and patients may be labeled properly, which may be quickly understood.

6.       Exercise reduces Abeta burden and oxidative stress, what should be the main target of exercise? It will be the anti-oxidative effect or the anti-Abeta?? This point may be clarified from the available literature.

7.       Overall, the paper has been nicely designed and efforts have been put into making it readable and understandable.

Comments on the Quality of English Language

English is fine. Minor editing is required

Author Response

Comments and Suggestions for Authors

The Paper entitled “Advancing Alzheimer's Therapeutics: Exploring the Impact of Physical Exercise in Animal Models and Patients by Jesús Andrade-Guerrero and co-authors highlighted the beneficial effects of exercise on Alzheimer’s disease-related pathologies in animals and humans. They have covered the underlying AD-related pathologies, such as oxidative stress, neuroinflammation, and cognitive functions. The paper has been strengthened with several tables and graphical illustrations. The overall flow of the paper is simple, stable, and nicely designed.

We thank the reviewer for the effort in analyzing our manuscript. We have highlighted in green the changes and suggestions made by the reviewer.

Q1.       The most part of the abstract is related to AD-pathology, it must present a high proportion of the exercise-inducing effects.

A1. We appreciate the reviewer's suggestion and have integrated the exercise-induced neuroprotective effects into the abstract.

Q2.       The given conclusion is insufficient, more research gaps may be targeted and highlighted, which may be a clue for future researchers.

A2. We thank the reviewer for the suggestions and we have added some gaps in the research focused on physical exercise in AD.

Q3.       The paper is quite lengthy, efforts may be made to summarize the main things and make it easier for the authors to make it publishable.

A3. We have tried to summarize some aspects in the different sections of the manuscript.

Q4.       Highlight the shortcomings or limitations of the current research on the role of exercise in preventing or managing AD-related pathologies, which may be added in the conclusion and future perspectives.

A4. We appreciate the reviewer's suggestion and we have added in the conclusion section the limitations of the current research on the role of exercise in preventing AD-related pathologies.

Q5.       Figure-1, effects of exercise on animal models and patients may be labeled properly, which may be quickly understood.

A5. We have followed the reviewer's suggestion.

Q6.       Exercise reduces Abeta burden and oxidative stress, what should be the main target of exercise? It will be the anti-oxidative effect or the anti-Abeta?? This point may be clarified from the available literature.

A6. According to the reviewer's suggestion, we have highlighted in the abstract and in the body of the manuscript that there is no main objective of the exercise but that the effect is multi-target.

Q7.       Overall, the paper has been nicely designed and efforts have been put into making it readable and understandable.

A7. We thank the reviewer for the comments and suggestions made to improve our manuscript.